# Fabrication of Activated Carbon Decorated with ZnO Nanorod-Based Electrodes for Desalination of Brackish Water Using Capacitive Deionization Technology

**DOI:** 10.3390/ijms24021409

**Published:** 2023-01-11

**Authors:** Jhonatan Martinez, Martín Colán, Ronald Castillón, Pierre G. Ramos, Robert Paria, Luis Sánchez, Juan M. Rodríguez

**Affiliations:** Center for the Development of Advanced Materials and Nanotechnology, Universidad Nacional de Ingeniería, Av. Túpac Amaru 210, Lima 15333, Peru

**Keywords:** zinc oxide, activated carbon, capacitive deionization, CDI, water desalination, salt absorption capacity, specific energy consumption, thermodynamic energy efficiency

## Abstract

Capacitive deionization (CDI) is a promising and cost-effective technology that is currently being widely explored for removing dissolved ions from saline water. This research developed materials based on activated carbon (AC) materials modified with zinc oxide (ZnO) nanorods and used them as high-performance CDI electrodes for water desalination. The as-prepared electrodes were characterized by cyclic voltammetry, and their physical properties were studied through SEM and XRD. ZnO-coated AC electrodes revealed a better specific absorption capacity (SAC) and an average salt adsorption rate (ASAR) compared to pristine AC, specifically with values of 123.66 mg/g and 5.06 mg/g/min, respectively. The desalination process was conducted using a 0.4 M sodium chloride (NaCl) solution with flow rates from 45 mL/min to 105 mL/min under an applied potential of 1.2 V. Furthermore, the energy efficiency of the desalination process, the specific energy consumption (SEC), and the maximum and minimum of the effluent solution concentration were quantified using thermodynamic energy efficiency (TEE). Finally, this work suggested that AC/ZnO material has the potential to be utilized as a CDI electrode for the desalination of saline water.

## 1. Introduction

Capacitive deionization (CDI) is a technology that, although it is not recent, has attracted the interest of many researchers in the last decade due to the great advance in the study of desalination processes employed mainly to solve freshwater shortage [1]. It is widely known that fresh water is a limited resource, so complex methods are often used to obtain it (e.g., electrodialysis, multi-effect distillation, and reverse osmosis), which requires high energy expenditure and periodic membrane replacement [2]. For this reason, devices designed under the concept of capacitive deionization are a great alternative due to their low energy consumption and the efficiency they provide [3,4].

The principle of capacitive deionization is centered on the temporary retention of ions in the inner part of electrodes, after which, the delivery to the water flow of the trapped ions is performed by reversing the electrodes’ electric polarity. For this reason, devices based on capacitive deionization should include in their description ion adsorption and desorption, which comprise a complete cycle and, depending on the electrode used, can perform a large number of cycles without losing any of their properties [5,6]. In general, the CDI cell performance can be affected by many factors including the wettability, electrical conductivity, and specific surface area of the active material [7,8]. In that regard, carbon based-materials have been considered ideal candidates for CDI cell electrodes to achieve all of these criteria [9,10,11]. Particularly, activated carbon (AC) is preferably used as an electrode material in the CDI process due to its scalability, cheapness, stability, and high specific surface area [12,13,14]. However, since commercial AC also exhibits a significantly disordered arrangement of micropores (less than 2 nm), its electrosorption capacity will be low [15]; thus, the whole desalination process will become inefficient.

Currently, to improve the CDI cell performance and overcome the disadvantages founded for AC electrodes, nanostructured metal oxides (MOs) have been applied as additives to carbon-based electrodes, achieving an improvement in the desalination performance [16,17]. For instance, ZnO, TiO_2_, SnO_2_, Fe_2_O_3_, and MnO_2_ have been widely employed as promising materials for desalination processes owing to their capability to store more electronic charge, high stability, and low toxicity [18,19,20,21]. Among the metal oxides mentioned above, nanostructured ZnO stands out from the other metal oxides due to its unique chemical and physical features, easy fabrication, lower cost, and high capacitance performance [22]. Furthermore, ZnO is a direct band gap semiconductor (3.37 eV), which can be easily synthesized and grown in various morphologies and shapes [23], such as nanoparticles, nanotubes, nanorods, among others [23,24,25]. These different types of nanostructures can enhance surface area and the double-layer capacitance of the carbon material, which results in a higher electrosorption capacity and, therefore, high desalination performance [26,27,28]. Thereby, in this study, using zinc oxide as an additive material, activated carbon/zinc oxide (ZnO/AC) composite electrodes were fabricated by simply mixing AC with ZnO nanostructures for subsequent testing as CDI electrodes in water desalination application. To evaluate the performance of CDI electrodes and compare it with other devices, the salt adsorption capacity (*SAC*), average salt adsorption rate (*ASAR*), specific energy consumption (*SEC*), Gibbs free energy of separation (*ΔG*), and thermodynamic energy efficiency (*TEE*) were calculated. *SAC* and *ASAR* directly involve the active mass of the electrode, whereas *SEC*, *ΔG*, and *TEE* provide information on the energy used during the deionization process. These obtained parameters will provide an important guide for the selection of materials in CDI processes.

## 2. Results and Discussions

### 2.1. SEM and DRX Characterization

The morphologies of the ZnO nanostructures, the activated carbon, and the AC/ZnO composite electrode are shown in Figure 1. Figure 1a displays an SEM image of the pristine ZnO nanostructures, which exhibits a mixture of rod-like and urchin-like nanostructures. The length and diameter of the ZnO nanostructures were measured using ImageJ software and SEM pictures of the nanostructures; the findings are presented in Appendix A. The average diameters and lengths have been determined to be 254 ± 12 nm and 3.17 ± 0.15 µm, respectively. Figure 1b,c show the surface morphology of the AC and AC/ZnO electrodes, respectively. As seen in Figure 1b, AC granules have both roughly structured surfaces and irregular sizes, which will facilitate the anchoring of the ZnO nanostructures on the AC surface. This phenomenon was confirmed in the SEM image for the AC/ZnO electrode (regions enclosed by white lines in Figure 1c). This event may facilitate the ionic diffusion and mass transfer along the electrode in the electrochemical cell, due to the semi-conductivity and hydrophilicity properties of ZnO [29] It is worth noting that only ZnO nanorods are observed on the surface of the AC electrode, which is likely due to the stirring and ultrasonication process of the slurry of the active material needed for the fabrication of the electrode results in the disintegration of urchin-like into rod-like nanostructures.

The crystallinity and phase purity of the AC/ZnO nanocomposites were characterized by X-ray diffraction analysis. The XRD patterns of the ZnO nanostructures, the activated carbon (AC), and the AC/ZnO composite are shown in Figure 2. The XRD pattern of the fabricated ZnO nanostructures contains six main diffraction peaks corresponding to the crystalline planes (100), (002), (101), (102), (110), and (103) (JCPDS 036-1451), ascribed to the hexagonal structure of ZnO wurtzite [30]. Moreover, the broad asymmetric diffraction peaks for AC were found at 2θ around 24° and 43°, which correspond to the (002) and (101) planes of crystalline hexagonal graphite (JCPDS 041-1487), respectively [31]. Finally, in the case of the AC/ZnO composite, its XRD diffractogram reveals the main characteristic peaks of zinc oxide, as well as a very weak and almost negligible peak corresponding to the (100) plane of AC, whereas the peak corresponding to (100) plane completely disappeared for the synthesized composite.

The variation in specific capacitance as a function of the amount of ZnO utilized to produce CA/ZnO electrodes and the thickness of the active layer is shown in Table 1. The values of specific capacitance were calculated according to Equation (7) and the cyclic voltammograms shown in the Appendix A. The results demonstrate that when the mass of ZnO utilized to fabricate the electrodes increases, the specific capacitance increases and the thickness of the electrodes decreases. This phenomenon can be described by the ion transportation theory, which establishes that as the thickness decreases, the transport of ions within the active layer is easier [31,32]. The highest value of specific capacitance calculated in this work was 135.7 F/g, obtained for the sample made with 9 g of ZnO (approximately 99 μm thickness), which is higher compared to that obtained for electrodes based only on activated carbon (69.4 F/g) calculated in our previous work [31]. The results show that the presence of ZnO nanorods in the matrix of CA electrodes increases its specific capacitance. The specific capacitance is used to predict the electrochemical performance of an electrode; a higher specific capacitance allows the retention of more ions in the charged double layer; therefore, a better efficiency can be obtained. Appendix A summarizes the BET analysis results of the active material of the AC and AC/ZnO (9 g) electrodes. It has been revealed that the specific surface area of the AC/ZnO composites is six times larger than that of pure AC.

### 2.2. Cell Performance

In this section, the performance of the CDI cell fabricated from nine pairs of electrodes of the highest specific capacitance obtained according to the results of Table 1 (electrode fabricated with 9 g of ZnO) was analyzed. The concentration in ppm of the ions obtained for multiple charge–discharge cycles is shown in Figure 3a. Three deionization cycles were carried out at a time interval of 5000 s. The obtained results reveal the typical charge–discharge behavior for a CDI cell, agreeing perfectly with what was observed in previous works [3,4]. In addition, all observed cycles have the same shape and have equivalent minimum and maximum values, resulting in equal cycle yields. The plot was obtained using 0.4 M of initial concentration, 1.2 V as voltage, and 85 mL/min as flow rate. This process was repeated for other flow rates between 45 mL/min and 105 mL/min, obtaining the same characteristic curve.

In the CDI field, the salt absorption capacity (*SAC*) is generally used to evaluate the capacity of the electrodes to remove salt. The *SAC* is obtained dividing the mass of salt removed (mg) by the mass of the electrode [33]. In the case of a continuous feeding system, the expelled water concentration is not constant over time, so an integral form of the *SAC* equation is used. In this way, *SAC* can be calculated according to Equation (1) [33], where *m* is the total active mass of the electrode (*g*) and *Q* is the feedwater flow rate (mL/min). Moreover, *C* and *C*_0_ represent the NaCl concentration of the ejected water at *t* (min) and the feed NaCl concentration, respectively.
(1)$$SAC\;\left(mg/g\right)=\frac{Q}{m}\int_{0}^{t}\left(C-C_{0}\right)dt$$

The average salt adsorption rate (*ASAR*) describes the salt removal capacity from the solution within a specific time. This value can be calculated by dividing the *SAC* value by the cycle time (Δ*t_cycle_*) or adsorption time (Δ*t_abs_*) using Equations (2) and (3), respectively [33,34].
(2)$$ASAR_{1}=\frac{SAC}{\Delta t_{cycle}}$$
(3)$$ASAR_{2}=\frac{SAC}{\Delta t_{abs}}$$

Figure 3b shows the comparison between the cycle time and adsorption time as a function of the flow rate used. The figure shows that the flow rate increased as both times decreased. These results will then be used to obtain the *ASAR*_1_ and *ASAR*_2_ values, respectively.

On the other hand, the variation of the *SAC* for different flow rates is shown in Figure 4a. The calculated *SAC* was found to be in the range of 93.14–123.66 mg/g approximately, with the value of 123.66 mg/g being the maximum obtained for a flow rate of 95 mL/min. Furthermore, the *ASAR*_1_ and *ASAR*_2_ as a function of the flow rate are shown in Figure 4b. The trend found for both plots is similar, with a maximum value of 5.06 mg/g/min (*ASAR*_1_) and 9.89 mg/g/min (*ASAR*_2_) for 95 mL/min of flow rate. In addition, at flow rates larger than 100 mL/min, the *SAC* and *ASAR* values declined significantly. These results are likely due to two causes. The first issue relates to the minimal ionic absorption produced before the stage of equilibrium is reached. The second relates to unsatisfactory CDI cell operation caused by the loss or separation of active material from the graphite sheet. Finally, the values obtained from *SAC*, *ASAR*_1_, and *ASAR*_2_ for different flow rates are presented in Table 2. The obtained results, shown in Table 2, confirm that the CDI cell fabricated in this work can operate at high concentrations of brackish water, which would allow us to work in more realistic conditions.

### 2.3. Specific Energy Consumption

Specific energy consumption in a CDI (*SEC*) cell describes the energy consumption per unit volume of water produced (Wh/m^3^). The *SEC* does not depend on the process or initial conditions and is determined from the following Equation (4) [35]:(4)$$SEC=\frac{1}{V_{D}}\int_{t_{o}}^{t_{d}}V\left(t\right)i\left(t\right)dt$$
where *i*(*t*) is the current, *V*(*t*) is the cell voltage, and *V_D_* is the volume of diluted water, at their initial time, while td is the absorption time.

In this research, three operating processes were analyzed, in which changes in polarity were made to force ions to be desorbed from the electrode surface and thus reducing or increasing the cycle time. In addition, various flow rates were used and the initial conditions were kept constant. The obtained data will provide information on the cell performance and the best-operating conditions. In the first of the processes, a polarity change was carried out when the CDI cell returned to the initial concentration, maintaining it until the desorption was finished. In the second process, the polarity change was made both at the equilibrium point (point of lowest concentration) and when the cell reached the initial concentration. Finally, in the third process, the source of voltage was disconnected at the equilibrium point until the initial conditions were reached, after which the polarity was changed.

Figure 5 shows the effect of the change in electric polarity for each process. It has been found that process 2 is the fastest one, with the shortest cycle time, and process 3 the slowest one. The time during which the source of voltage was turned on and energy used for different flow rates in each procedure are shown in Table 3. The results reveal that the energy consumption for the third process (Figure 5c) was the lowest one in comparison to the two others; this is a predictable result given that the voltage was shut off throughout the desorption stage in this process. However, as indicated above, this process requires additional time than processes 1 and 2 to complete an absorption–desorption cycle. Therefore, time of operation is a necessary parameter to be considered in performance comparisons between the operations processes.

The specific energy consumption (*SEC*) for each process is shown in Table 4. The *SEC* values obtained for equal flows are roughly similar. Moreover, Figure 6 shows the variation of the average time as a function of the different flows used for each part of the deionization process. First, the shortest stage of the cycle shown in Figure 6a represents the linear ionic absorption of the electrodes until reaching ionic equilibrium, taking into account that at this point it can take a few seconds. Figure 6b shows the loading stage of the cycle which is the longest compared to the other stages. From Figure 6b, it can be seen that the increase in flow generates a decrease in time. Finally, Figure 6c shows the last stage of the cycle which is the discharge stage. This process exhibits the same trend seen in Figure 6b.

The time and energy required to complete a deionization cycle for the three procedures previously described are shown in Figure 7a,b, respectively. As can be seen from both plots, the energy consumption is higher for processes 1 and 2 since, due to the given conditions, the CDI cell is always powered. However, in the case of process 3, the energy consumption is the lowest one since the movement of the forced ions is due only to the initial charge and solution flow.

On the other hand, the CDI cell performance for different processes and flow rates was evaluated using the *SEC*. The obtained results are shown in Figure 7c as well as in Table 4. It should be noted that all the implemented procedures have similar *SEC* values and a decrement in the *SEC* with an increment in flow rates is observed.

### 2.4. Gibbs Free Energy of Separation

In order to determine the energy consumption used during the deionization process as such, the Gibbs free energy of separation (Δ*G*) was calculated. This parameter makes it possible to know the energy used to remove ions from the active mass of the electrode, since part of the energy supplied to the CDI cell is lost due to processes inside it. The value of Δ*G* can be quantified by Equation (5) [36]:(5)$$\Delta G=2RT\left\{\frac{C_{0}}{\gamma }ln\left[\frac{C_{0}-\gamma C_{D}}{C_{0}(1-\gamma }\right]-C_{D}ln\left[\frac{C_{0}-\gamma C_{D}}{C_{D}(1-\gamma }\right]\right\}$$
where *C*_0_ is the initial concentration, *C_B_* is the concentration of ionic equilibrium, *C_D_* is the maximum concentration of the desorption stage, *T* is absolute temperature, *R* is the Boltzmann constant, and *γ* is defined as the fraction of feedwater volume and the deionized water recovered, calculated by the following Equation (6) [36,37]:(6)$$\gamma =\frac{C_{0}-C_{B}}{C_{D}-C_{B}}$$

Figure 8a depicts the Gibbs free energy (*G*) as a function of various flow rates for the three processes examined in this study. The figure reveals that the Gibbs free energy of separation required to remove ions from the active mass of the electrode is the lowest for flow rates between 70 and 90 mL/min for the three processes. In addition, the Gibbs free energies calculated for different flow rates and processes from the data obtained from a deionization cycle, keeping the initial concentration and the voltage source constant, are shown in Table 5.

On the other hand, the energy fraction used to separate ions was determined from *ΔG* and *SEC*, which is defined as the thermodynamic energy efficiency (*TEE*) [37]. The results are shown in Figure 8b and Table 5. The *TEE* values calculated for the three types of processes are in the 0.05–0.09 range for flow rates in the 45–85 mL/min range.

## 3. Materials and Methods

### 3.1. Materials

All reagents used in the experiments were of analytical grade and used without any further purification. Zinc nitrate hexahydrate (Zn(NO_3_)_2_ 6H_2_O, Sigma-Aldrich, Steinheim, Germany) and sodium hydroxide (NaOH, Merck, Darmstadt, Germany) were used for the ZnO nanorods synthesis. Meanwhile, activated carbon (CAS 7440-44-0, M_w_~12.01, Sigma-Aldrich Co., Burlington, MA, USA), polyvinyl alcohol (CAS 9002-89-5 Mowiol, 10–98/M_w_~61,000, Sigma-Aldrich Co., Burlington, MA, USA), and glutaric acid (CAS 110-94-1, M_w_~132.11, Merck, Burlington, MA, USA) were used to prepare the active material.

Commercial graphite sheets were used as substrate for the electrodes. The electrochemistry and desalination tests were conducted using sodium chloride 0.2 M (NaCl anhydrous, ≥99% purity, M_w_~58.44.11, Sigma-Aldrich, Burlington, MA, USA).

### 3.2. Synthesis of ZnO Nanostructures

ZnO nanostructures were obtained by the hydrothermal process according to Wirunmogkol et al. [38] by mixing under vigorous stirring equal volumes of zinc nitrate hexahydrate (0.15 M) and sodium hydroxide (2.1 M) in water for 12 h at a constant temperature of 80 °C. Then, the formed precipitate was filtered and finally, the resulting powder was dried at 60 °C for 1 h.

### 3.3. Fabrication of the Electrode

Activated carbon/zinc oxide (AC/ZnO) electrodes were fabricated using a slurry of the active material, which was prepared as a suspension of activated carbon powder (45 g), polyvinyl alcohol (9 g), glutaric acid (15 g), and different amounts of the as-synthesized ZnO powders (3, 5, 7 and 9 g) in 150 mL of water. The mixture was stirred for 2 h and sonicated for 40 min to ensure homogeneity. After that, the slurry was deposited on a graphite sheet by using the doctor blade method with the help of a film coater (MIT MSK-AFA-III-HB) and a micrometer adjustable film applicator (MIT EQ-Se-KTQ-100), which allowed us to modify the initial thickness of the wet film. Finally, the coated graphite sheets were dried at room temperature for 2 h and then placed in an oven at 120 °C for one hour.

### 3.4. Fabrication of CDI Cell

The CDI cell consists of nine pairs of electrodes, one of these pairs is illustrated in Figure 9a. An electrode consists of two elementary components, the substrate, which in this case is a graphite sheet of 120 mm × 120 mm × 1 mm dimensions, and the AC/ZnO active material, which covers the central area of the graphite sheet of 100 mm × 100 mm dimensions. A blind rubber gasket of 120 mm × 120 mm × 1mm dimensions was used to separate two electrodes with their active layers facing each other. The blind rubber gasket has a 100 mm × 100 mm central aperture through which the saline solution can circulate. Finally, to create the CDI cell, nine pairs of electrodes were arranged side by side with an insulator in between them to prevent short circuits. A schematic of the fabricated CDI cell is depicted in Appendix A.

### 3.5. Electrochemical Setup

A schematic of the complete CDI system used for the electrochemical measurements is shown in Figure 9b. This consists of a 15 L container used to store a 0.4 M NaCl solution, which was pumped into the CDI cell by using a peristaltic pump, model MasterFlex L/S Digital Dispensing Pump Drives. A 1.2 V DC source (PellTron 3005D) was used to energize the cell; meanwhile, the ion concentration was measured by using the multi-parameter device HANNA HI 5522.

### 3.6. Characterization Techniques

The morphology of the AC/ZnO electrode obtained was visualized by using scanning electron microscopy (SEM, Zeiss EVO-10). The structural properties were studied by using X-ray diffraction (diffractometer DRX Bruker D8 Advance). The electrochemical properties of the as-prepared electrodes were examined by cyclic voltammetry (CV). CV tests were performed using a three-electrode cell, where the AC/ZnO based-material was used as a working electrode, a platinum wire as a counter electrode, the Ag/AgCl electrode immersed in 0.1 M KCl solution as a reference electrode, and 0.5 M NaCl as an electrolytic solution. Voltammetry measurements were performed with a potentiostat (Metrohm Autolab PGSTAT204) at an operating window from 0.0 to 1.2 V vs. ref. (0.1 V) in a 0.5 M NaCl electrolyte. The specific capacitance *C* (F/g) was determined considering the mass (g) of the active material on the electrode surface (1 cm^2^) using Equation (7) [39]:(7)$$C_{s}=\frac{\int idV}{2v\;\Delta V\;m}$$
where *C_s_* is the specific capacitance (F/g), *i* is the measured electric current at different voltages (A), *V* is the applied voltage (V), *v* is the scan rate (V/s), and *m* is the mass of active material on the electrode (mg).

### 3.7. CDI Measurements

The desalination performance was obtained using the CDI setup described in Section 2.4. The ion adsorption and desorption steps were performed using the constant potential mode at 1.2 V, while electrode regeneration was achieved by reversing the applied voltage to 1.2 V. All experiments were carried out with an initial concentration of 0.4 M NaCl solution and a 15 L electrolyte tank. Furthermore, the influence of the operating conditions, the mass of ZnO used to fabricate CA/ZnO electrodes, as well as their thicknesses on the CDI system performance were investigated. Flow rates from 45 to 105 mL min^−1^ with steps of 10 mL min^−1^ were analyzed. Average salt adsorption rate (*SAR*) and salt adsorption capacity (*SAC*) were calculated from recorded ion concentration data, which are values commonly used in CDI systems research.

## 4. Conclusions

In summary, composites based on ZnO nanorods incorporated into activated carbon were developed and studied as an electrode for CDI. The results show an effective formation of AC/ZnO electrodes, where the adhered to ZnO nanorods significantly improves the electrochemical performance of the fabricated AC/ZnO composite. It was found that the AC/ZnO electrodes produced from 9 g of ZnO showed the highest value of specific capacitance (135.7 F/g) compared with the pristine AC composite, and therefore they were utilized to fabricate and analyze the performance of the CDI cell. In addition, the salt adsorption capacity (*SAC*), average salt adsorption rate (*ASAR*), specific energy consumption (*SEC*), Gibbs free energy of separation (Δ*G*), and thermodynamic energy efficiency (*TEE*) were calculated obtaining the highest values of these indicators for a flow rate of 95 mL/min. Finally, this research demonstrates the potential of the AC/ZnO composite used in the CDI system to enhance brackish water desalination and proposes a particularly important base and guidance for the optimization of the capacitive desalination process.

## Figures and Tables

**Figure 1 ijms-24-01409-f001:**
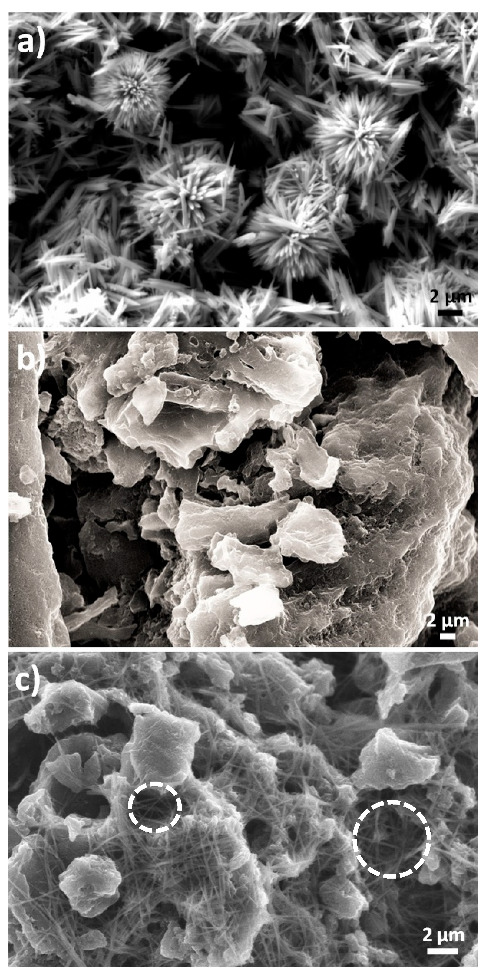
SEM images of (**a**) ZnO nanostructures, (**b**) AC, and (**c**) AC/ZnO composite.

**Figure 2 ijms-24-01409-f002:**
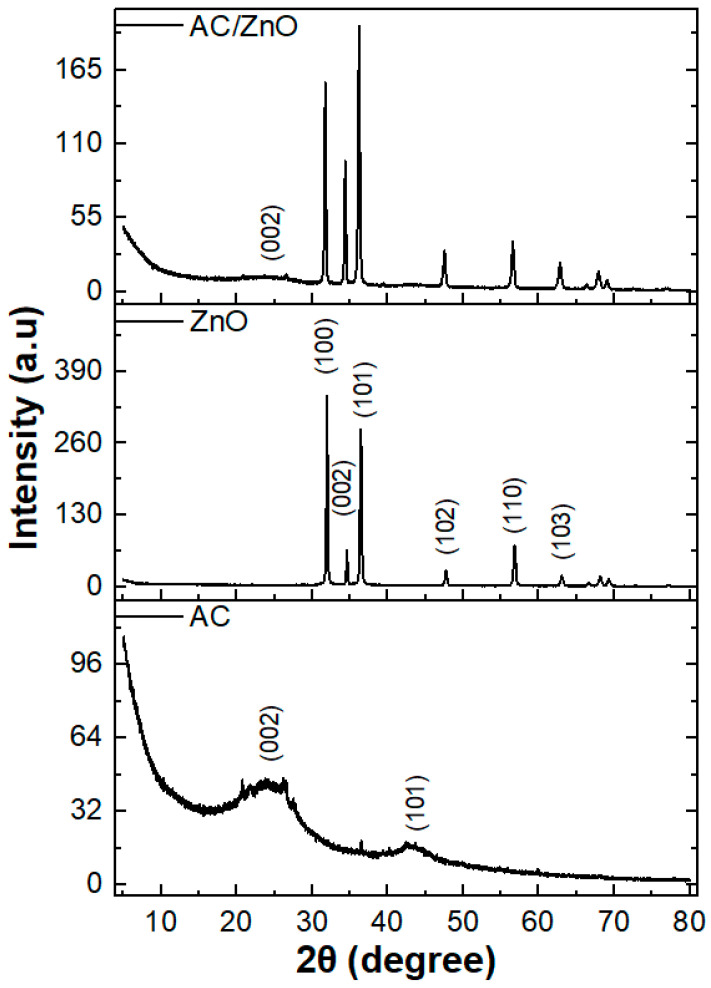
XRD patterns of ZnO nanostructures, AC, and AC/ZnO composite.

**Figure 3 ijms-24-01409-f003:**
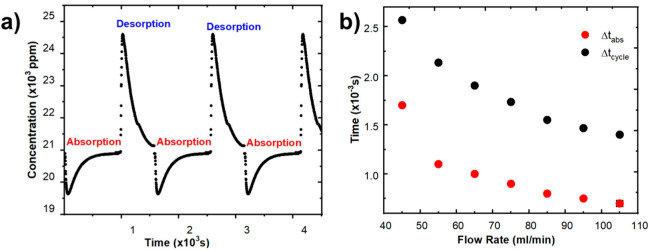
(**a**) Deionization cycles (charge–discharge) of CDI cell (85 mL/min, NaCl 0.4 M, 1.2 V). (**b**) Comparison between cycle time (Δ*t_cycle_*) and adsorption time (Δ*t_abs_*) as a function of flow rate (NaCl 0.4 M, 1.2 V).

**Figure 4 ijms-24-01409-f004:**
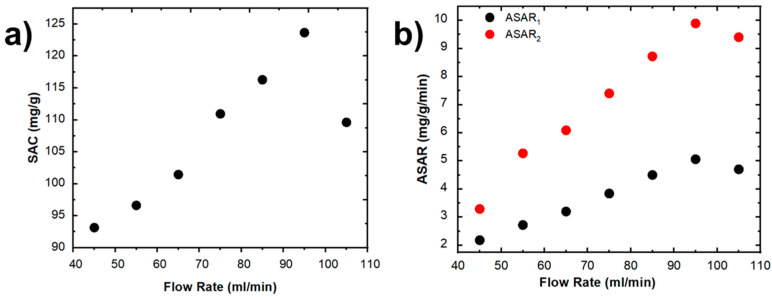
Variation of (**a**) *SAR* and (**b**) *ASAR* as a function of flow rate (NaCl 0.4 M, 1.2 V).

**Figure 5 ijms-24-01409-f005:**
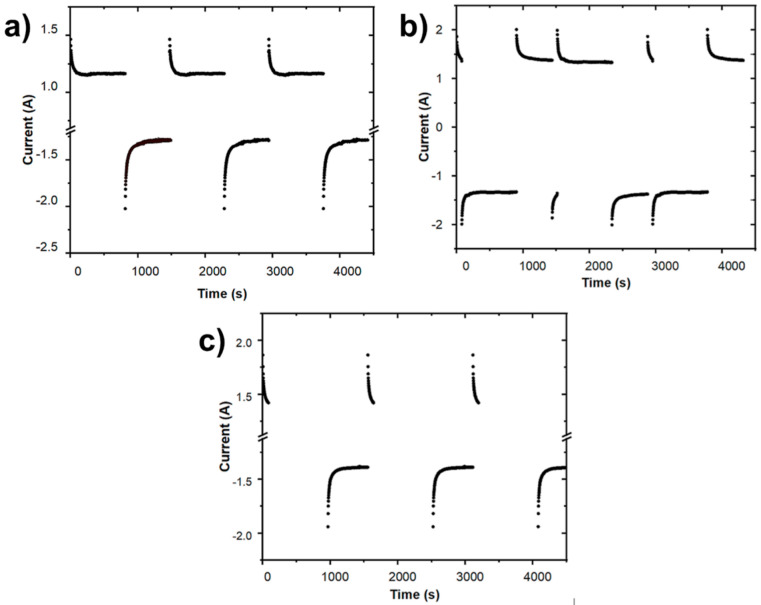
Current versus time curve of absorption–desorption cycles of the CDI (NaCl 0.4 M, 1.2 V, 75 mL/min) for three types of processes: (**a**) Process 1, (**b**) Process 2, and (**c**) Process 3.

**Figure 6 ijms-24-01409-f006:**
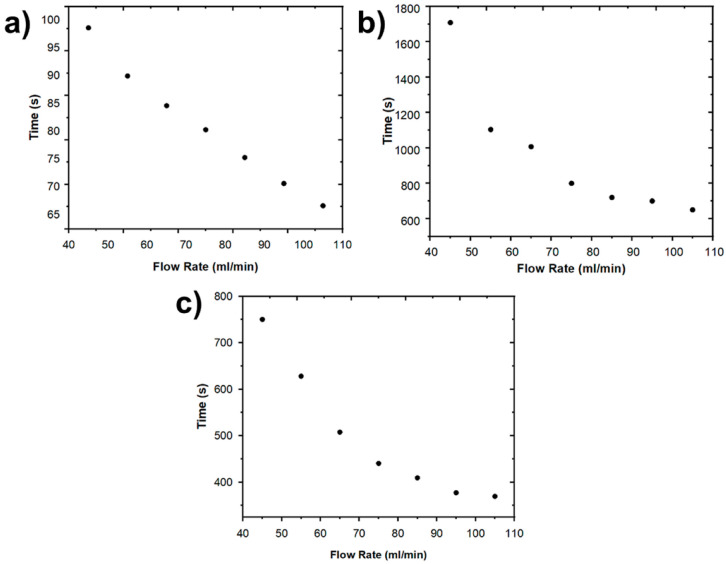
Time as a function of flow rate (NaCl 0.4 M, 1.2 V) to reach: (**a**) The ionic equilibrium (lower concentration), (**b**) the initial concentration (charge stage), and (**c**) the initial concentration after discharge stage.

**Figure 7 ijms-24-01409-f007:**
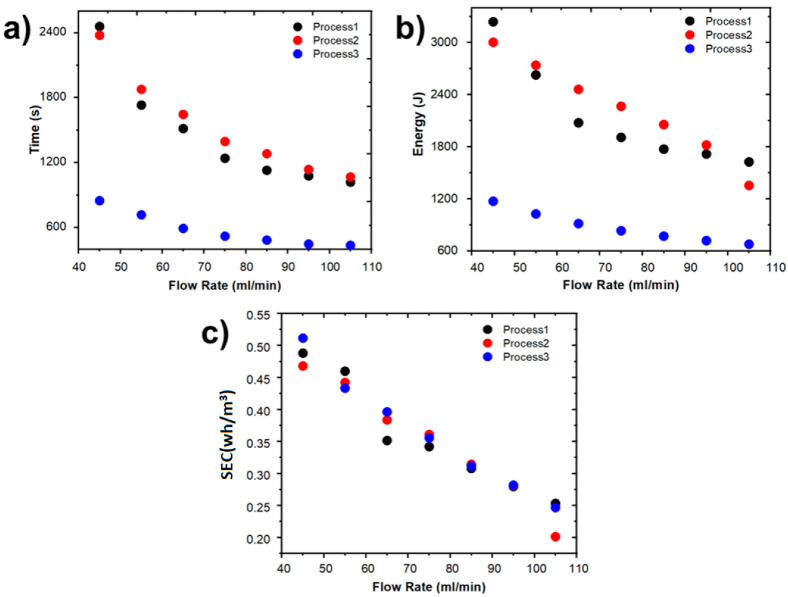
(**a**) Time, (**b**) energy, and (**c**) *SEC* for the three types of processes as a function of flow rate (NaCl 0.4 M, 1.2 V).

**Figure 8 ijms-24-01409-f008:**
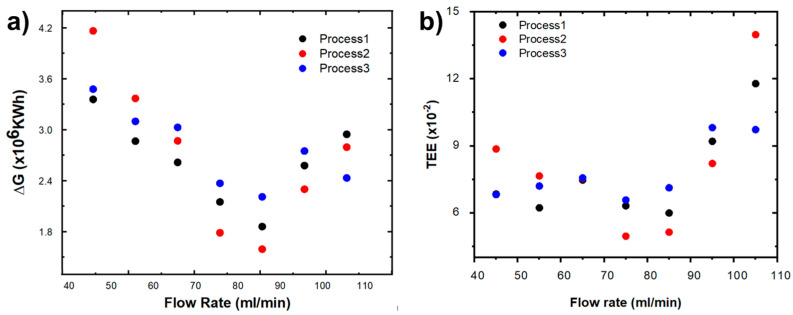
(**a**) Gibbs free energy and (**b**) Thermodynamic energy efficiency for the three types of processes as a function of flow rate (NaCl 0.4 M, 1.2 V).

**Figure 9 ijms-24-01409-f009:**
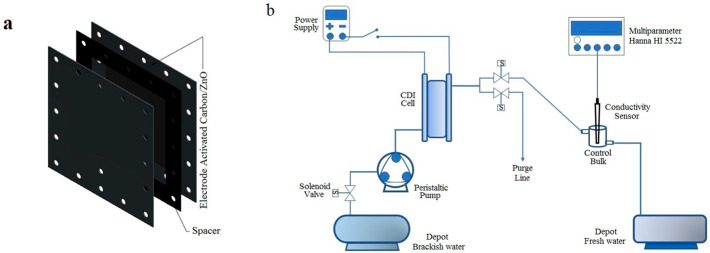
(**a**) Schematic of one of the nine pairs of electrodes of a CDI cell and (**b**) schematic of the main experimental setup used for the CDI studies. In this diagram, different elements such as CDI cell, peristaltic pump, DC power supply, conductivity sensor, and pH meter are displayed.

**Table 1 ijms-24-01409-t001:** Measured thickness and specific capacitance for various electrodes fabricated with different amounts of ZnO.

ZnO Mass (g)	Measured Thickness (μm)	Specific Capacitance (F/g)
3	104	67.6
5	102	109.4
7	101	124.7
9	99	135.7

**Table 2 ijms-24-01409-t002:** Comparison of *SAC* and two types of *ASAR* calculated for various flow rates (NaCl 0.4 M, 1.2 V).

Flow Rate	SAC	ASAR_1_	ASAR_2_
(mL/min)	(mg/g)	mg/g/min	mg/g/min
45	93.14	2.18	3.29
55	96.63	2.72	5.27
65	101.45	3.20	6.09
75	110.95	3.84	7.40
85	116.29	4.50	8.72
95	123.66	5.06	9.89
105	109.62	4.70	9.40

**Table 3 ijms-24-01409-t003:** Time and energy consumed for various flow rates using three different types of processes (NaCl 0.4 M, 1.2 V).

Flow Rate	Process 1		Process 2		Process 3	
	Time	Energy Consumed	Time	Energy Consumed	Time	Energy Consumed
(mL/min)	(s)	(J)	(s)	(J)	(s)	(J)
45	2458	3239.1	2377	3003.4	849	1172.6
55	1732	2628.0	1877	2739.2	718	1026.6
65	1515	2076.2	1645	2461.2	592	915.5
75	1240	1908.2	1396	2266.1	521	833.0
85	1129	1772.7	1283	2056.1	485	770.2
95	1077	1716.3	1137	1820.9	448	719.4
105	1020	1625.5	1068	1354.3	436	677.8

**Table 4 ijms-24-01409-t004:** Specific energy consumption (*SEC*) variation as a function of flow rate for the three sorts of the process (NaCl 0.4 M, 1.2 V).

Flow Rate (mL/min)	Process 1	Process 2	Process 3
	SEC		
	(Wh/m^3^)	(Wh/m^3^)	(Wh/m^3^)
45	0.49	0.47	0.51
55	0.46	0.44	0.43
65	0.35	0.38	0.40
75	0.34	0.36	0.36
85	0.31	0.31	0.31
95	0.28	0.28	0.28
105	0.25	0.20	0.25

**Table 5 ijms-24-01409-t005:** Gibbs free energy of separation and thermodynamic energy efficiency calculated for various flow rates using three different types of processes (NaCl 0.4 M, 1.2 V).

Flow Rate (mL/min)	Process 1		Process 2		Process 3	
	ΔG (×10^−6^ KWh)	TEE (×10^−2^)	ΔG (×10^−6^ KWh)	TEE (×10^−2^)	ΔG (×10^−6^ KWh)	TEE (×10^−2^)
45	3.36	6.85	4.17	8.87	3.48	6.82
55	2.87	6.23	3.37	7.66	3.10	7.21
65	2.62	7.48	2.87	7.55	3.03	7.57
75	2.15	6.32	1.79	4.96	2.37	6.58
85	1.86	6.00	1.59	5.14	2.21	7.13
95	2.58	9.21	2.30	8.21	2.75	9.82
105	2.95	11.79	2.80	13.98	2.43	9.73

## Data Availability

Not applicable.

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
