# Peer review of "Fabrication of Activated Carbon Decorated with ZnO Nanorod-Based Electrodes for Desalination of Brackish Water Using Capacitive Deionization Technology"

_ijms, 2023, doi:10.3390/ijms24021409_

Round 1
Reviewer 1 Report
Track changes mode is turned on in some places and needs to be fixed. Apart from that, there is no harm in publishing it as it is. In some places, there are grammatical errors and spell check is required.
Author Response
We appreciate the reviewer's possitive comments
Reviewer 2 Report
The submitted work reports on the fabrication and characterization of composites based on ZnO nanorods incorporated into activated carbon as an electrode for CDI. Results show an effective formation of ZnO/AC electrodes, where the adhered of ZnO nanorods notably improves the electrochemical performance of the fabricated ZnO/AC composite. The work reveals interesting results that might be a subject of interest to the readers especially for high-performance CDI electrodes for water desalination. However, several comments/issues are proposed herein for further improvement of the scientific quality of the submitted manuscript before it can be accepted for publication.
1. Water desalination or water desalinization (in keyword)? Please use only one specific term for consistency through out the manuscript.
2. Content in page 3 are heavily re-used from previous report by the authors (Martinez J, Colán M, Catillón R, Huamán J, Paria R, Sánchez L, Rodríguez JM. Desalination Using the Capacitive Deionization Technology with Graphite/AC Electrodes: Effect of the Flow Rate and Electrode Thickness. Membranes. 2022; 12(7):717. https://doi.org/10.3390/membranes12070717). Suggest to rephrase/revise and cite previous work where necessary to avoid any inconveniences.
3. In line 148, Figure 1a should be Figure 2a since it's referring to SEM image.
4. Based on XRD profiles of the ZnO/AC composite, the broad peaks corresponding to AC located at the 2θ values of 43.6° is not really visible. Please justify.
5. In line 181, '....(approximately 99 um thickness),,' please use the correct way of writing the micrometer.
6. From line 183-186: 'The results obtained show that the ZnO with activated carbon mixture for its subsequent use in the CDI electrodes fabrication, will increase its specific capacitance and therefore electrodes with greater electrochemical efficiency will be obtained, retaining a greater number of ions in the charged double layer.' How did the authors draw this conclusion?
7. In line 242, 'Finally, in the third method..' Please replace 'method' with 'process' for consistency.
8. Based on Figure 6, please explain how polarization is affecting the proposed operating processes?
9. Are you suggesting that the energy consumption for Process 3 will be at par with the other two processes if more time is acquired? Justify
10. In line 278-279, 'Furthermore, for flow rates greater than 110 mL/min two problems arise.' Any related results to support this?
11. In line 291, 'CB is the concentration of ionic equilibrium', I don't see any CB in equation 6.
12. From line 296-298, 'From the figure, it is obtained that the 296 optimal flow rates were found in the 70-90 mL/min range because the Gibbs free energy of separation necessary to remove the ions from the active mass of the electrode is lower in the 3 contemplated processes.' How did authors come out with this conclusion?
13. Suggest to include the cyclic voltammograms in the main text with explanation.
14. Some units are not correctly written throughout the manuscript (subscript/superscript). Please check overall paper and rectify.
15. What are the benefits and limitations of these type of composites in term of real application?
Author Response
Many thanks for the reviewer's comments, our response is attached

Reviewer 3 Report
This manuscript discusses the synthesis of activated carbon decorated with ZnO nanorods based electrodes for capacitive deionization application.
Comments
1- The introduction of the manuscript needs to be rewritten in a better way, as there are many weaknesses unclear words such as (new proposals that have been presented) what do you mean by proposals?
2- I don’t see any novelty, could you please let me know what s the novelty?
3- How much AC and ZnO have you used?
4- What is the size of the ZnO?
5- Three electrosorption and regeneration cycles is not enough to show the cyclic stability, please run at least 10 cycles.
6- Please enrich the manuscript with relevant work published in the journal such as
https://doi.org/10.1039/C9EW01033E
10.1088/2053-1583/ab2927
7- Can you explain more about how the CDI cell consists of nine pairs of electrodes
8- Give the calculation formula for the CV-specific capacitance
9- The authors should provide the detailed parameters and process of electrochemical characterization
- The authors should provide the surface area and pore size volume for the material
1- After electrosorption, the CDI electrodes should be measured by XRD, SEM, TEM, and BET, etc.
Author Response

(The authors gave the same response as above.)

Round 2
Reviewer 2 Report
Authors have sufficiently addressed all the comments and suggestion as presented in the revised version of the manuscript.
But only one minor changes needed (based on my previous comment no.11), referring to line 342 (revised version), I believe authors imply that CB is referred from equation 6 but the actual case is it appears in equation 7. Suggest to rephrase/revise sentence accordingly to avoid misunderstanding.
Author Response
All comments were taken into account
Reviewer 3 Report
I have no more comments
Author Response
All comments were taken into account